# Anastasis and Other Apoptosis-Related Prosurvival Pathways Call for a Paradigm Shift in Oncology: Significance of Deintensification in Treating Solid Tumors

**DOI:** 10.3390/ijms26051881

**Published:** 2025-02-22

**Authors:** Razmik Mirzayans

**Affiliations:** Department of Oncology, University of Alberta, Cross Cancer Institute, Edmonton, AB T6G 1Z2, Canada; razmik.mirzayans@ahs.ca

**Keywords:** intratumor heterogeneity, therapy resistance, minimal residual disease, apoptosis, anastasis, PGCCs, genome chaos, CD24

## Abstract

What is apoptosis? The Nomenclature Committee on Cell Death and numerous other pioneering cancer/p53 biologists use the terms “apoptosis” and “cell death” interchangeably, disregard the mind-numbing complexity and heterogeneity that exists within a tumor (intratumor heterogeneity), disregard the contribution of polyploid giant cancer cells (PGCCs; the root causes of therapy resistance and relapse) to this heterogeneity, and then propose novel apoptosis-stimulating anticancer strategies. This is shocking for the following three reasons. First, clinical studies reported since the 1990s have revealed that increased apoptosis in solid tumors is associated with increased tumor diversity and poor prognosis. Second, we have known for years that dying (apoptotic) cancer cells release a panel of secretions (e.g., via phoenix rising and other pathways) that promote metastatic outgrowth. Third, over a decade ago, it was demonstrated that cancer cells can recover from late stages of apoptosis (after the formation of apoptotic bodies) via the homeostatic process of anastasis, resulting in the emergence of aggressive variants. The cell surface expression of CD24 has recently been reported to be preferentially enriched in recovered (anastatic) cancer cells that exhibit tumorigenic properties. These and related discoveries outlined herein call for a paradigm shift in oncology to focus on strategies that minimize the occurrence of treacherous apoptosis and other tumor-repopulating events (e.g., therapy-induced cancer cell dormancy and reactivation). They also raise an intriguing question: is deregulated anastasis (rather than evasion of apoptosis) a hallmark of cancer?

## 1. Introduction

Spontaneous cancer remission occurs in a small fraction of patients with solid tumors after an invasive biopsy procedure or infection with various pathogens [1]. There are also rare cases where conventional therapies result in long-term remission, and sometimes even cancer cure. This small fraction of patients are referred to as “exceptional responders” [2]. For the majority of cancer patients, however, intensive anticancer treatment which is designed to eradicate solid tumors has proven to cause more harm than benefit. In fact, the life expectancy of patients with some malignancies (e.g., esophageal cancers) has not improved significantly over the span of a century [3]. This is perhaps not surprising, given that a presumed friend (apoptosis) has turned out to be the worst enemy in cancer therapy [4,5,6,7,8,9,10,11,12,13,14,15,16,17].

Engaging regulated cell death via apoptosis in cancer cells has long been viewed as the ultimate goal in achieving remission and favorable therapeutic outcomes. However, numerous studies published over the past three decades have demonstrated that increased apoptosis in situ correlates with tumor aggressiveness and poor prognosis (reviewed in, e.g., [18]). The pro-oncogenic activity of apoptosis has been demonstrated in clinical studies for colorectal carcinoma [19,20,21,22,23,24,25], breast carcinoma [26,27,28], gastric carcinoma [29], malignant mesothelioma [30,31], non-small-cell lung cancer [32,33], pancreatic duct carcinoma [34,35,36], and squamous carcinoma of the tongue [18], as well as in preclinical (e.g., therapeutic mouse model) studies for other malignancies (see Table 1 in [18]). These observations do not support the popular hypothesis that apoptosis might be a critical tumor suppression mechanism.

Although the dark side of apoptosis in cancer therapy has been known since the 1990s [25,29,32,33], the majority of publications (articles, reviews, editorials, online blogs) that discuss the challenges and opportunities of cancer therapy disregard this well-established reality and propose novel apoptosis-stimulating anticancer strategies. These include recent publications by the Nomenclature Committee on Cell Death (NCCD) [37] and other pioneering authors in high-impact-factor journals (e.g., [38,39,40,41,42,43,44,45,46,47,48,49,50]). What are the reasons for this serious oversight? Who knows! Perhaps “in the quest for the next cancer cure, few researchers bother to look back at the graveyard of failed medicines to figure out what went wrong” [51].

Mustapha Kandouz [52] has recently published a comprehensive review on the impact of intercellular communication in cell fate outcomes after engaging apoptosis, in which he has raised a fundamental question: “What is cell death, really?”. The intention of the current commentary is to shed some light on this question with a focus on apoptosis. Namely, what is apoptosis and when is it not accompanied by a prosurvival pathway?

Specifically, the objective of this article is twofold. First, to outline studies that support the emerging scenario illustrated in Figure 1 (Section 2, Section 3 and Section 4). The take-home messages include the significance of deintensification to minimize the occurrence of apoptosis and other tumor-repopulating events. Second, to highlight false hypotheses that have hampered progress in cancer research for decades and unfortunately continue to do so (Section 5 and Section 6).

Please note that we follow the concept of “one shoe does not fit all” in terms of therapy-induced responses in different cell types. Thus, as in our previous publications (e.g., [3,15,53]), the current commentary is focused on solid tumors and tumor-derived cell lines; the conclusions drawn may or may not pertain to other malignancies. In addition, we use quotation marks for vague terms such as “viability” and “lethality” for reasons discussed by us [3,15,53] and others [10,54,55] (also see below).

## 2. Growing Complexity of Cancer Cell Response to Therapeutic Agents: Beyond Repair, Arrest, and Suicide (Apoptosis)

The main purpose of the various Special Issues of MDPI publications that I have Guest Edited in the past five years has been to provide a comprehensive update on the growing complexity and heterogeneity that exists within a solid tumor (intratumor heterogeneity), with different subpopulations contributing to disease recurrence following anticancer treatment (Figure 2). Most articles published in these collections focused on therapy resistance reflecting regulated cell death (apoptosis), cancer cell dormancy (durable proliferation arrest), and cell fusion [53,56].

In 2021, Boris Zhivotovsky (member of the NCCD) and his associates (Zaitceva et al. [10]) published a comprehensive review in one of these Special Issues discussing the clinical implications of anastasis, the phenomenon that rescues cancer cells from late stages of apoptosis, and other modes of regulated cell death. The identification of this homeostatic cell recovery process indicates “the importance of developing a new direction in the study of tumor resistance…Anastasis challenges cancer treatment as it not only allows cells to survive, but also favors their further malignancy” [10].

Zaitceva et al. [10] made the following remarks with respect to assessing cancer cell fate: “Apart from cell visualization, the most widely used methods for the detection of cell death are western blotting, flow cytometry, and the viability assay (methods based on assaying of ongoing cellular metabolism and enzyme activity, such as MTT) which reflects summarized and averaged response of a large number of cells. A significant disadvantage of the cell population-based analysis is obscuring the presence of functionally important subpopulations in tumors”.

These subpopulations include anastatic cancer cells, as discussed by these authors, as well as polyploid/multinucleated giant cancer cells (PGCCs) [57], senescent cancer cells [58], radiation-tolerant persister cancer cells [59], drug-tolerant persister cancer cells [60], and quiescent cancer cells [61]. The responses denoted by asterisks in Figure 2 contribute to cancer cell dormancy (also called “senescence-like dormancy” and “minimal residual disease”) post-treatment. The importance of each of these responses to therapy resistance and disease recurrence has recently been discussed (e.g., [58,59,60,61]).

The limitations of preclinical assays that are ubiquitously used to identify novel anticancer drugs and treatment strategies have also been discussed [3,10,15,53,54,55]. These include high-content multiwell plate cell “viability” assays, as noted by Zaitceva et al. [10], as well as the in vitro colony formation assay, immunoblot/flow cytometric measurement of apoptosis and other regulated cell death pathways, and tumor growth studies in live animals [15,53]. These so-called “down” assays (decreased gene expression, decreased proliferation, decreased tumor growth, etc.) either overlook therapy-induced cancer cell dormancy (seeds for recurrence and metastasis) or score them “dead” [15,53].

In short, one of the purposes of these Special Issues has been to bring into focus an established reality: the majority (thousands) of cancer therapy-related articles that rely on cell “viability” and other preclinical “down” assays often report dangerously misleading and highly biased science fiction rather than science of any clinical relevance. These studies not only disregard cellular complexity and heterogeneity within a single tumor (including the responses noted in Figure 2) but also the impact of the immune system on the outcome of anticancer treatment. It is therefore not surprising that “The War on Cancer Isn’t Won Yet” [62] after half a century of extensive research and numerous clinical trials.

## 3. The Apoptosis–Anastasis Survival Pathway Amounts to a Paradigm Shift in Oncology

### 3.1. The Illusion of a Point of No Return After Engaging Apoptosis

In 2009, the NCCD [63] published a cautionary article formulating several caveats concerning the misuse of terminologies (e.g., cell survival, cell death, apoptosis, necrosis, autophagy) and concepts that had slowed down progress in the area of cell death research. Specifically, the NCCD invited the authors, reviewers, and editors of scientific periodicals to abandon the use of vague expressions such as ‘percentage apoptosis’ and to replace them with the cellular, molecular, and biochemical parameters that are actually measured. The NCCD also stressed the importance of distinguishing dead cells and dying cells that can recover and proliferate. Dying cells were defined as cells that engage in a regulated cell death process that is reversible until a ‘point-of-no-return’ is trespassed. For cells undergoing apoptosis, the NCCD concluded that this irreversible phase must be downstream of massive caspase activation, complete permeabilization of the mitochondrial outer membrane, and exposure of phosphatidylserine residues that emit ‘eat me’ signals for phagocytes. Thus, the concept of a restriction point for apoptotic cell death remained to be specifically defined.

In the same year, Hogan Tang and associates [64] reported that apoptosis is reversible in various cancer cell lines and that this reversal occurs after the activation of caspase cascade, excessive mitochondrial fragmentation, nuclear condensation, and cell shrinkage. As noted in an online publication [65], Tang subsequently joined the laboratory of Denise Montell of the University of California, Santa Barbara, where the phenomenon of recovery from the brink of death was further studied in the context of development, health (homeostasis), and disease. It was first named anastasis in 2012 [4].

Figure 3 shows the morphology of cancer cells undergoing the apoptosis–anastasis survival pathway (the three images are screenshots of a video presented online by Montell [66]). The following two points should be noted. First, anastasis occurs in virtually all apoptotic cells, including cells with apoptotic bodies. Second, anastasis is observed after clinically relevant anticancer exposure. Namely, the incubation of cells with a moderate concentration of an apoptosis stimulus (e.g., a chemotherapeutic drug) for a relatively short period of time (~24 h), followed by washing away the stimulus. On the other hand, as cautioned by Eastman [54], the continuous incubation of cells with high concentrations of anticancer agents that is typically used in cell death-related studies (apoptosis, flow cytometry, cell “viability” assays, etc.) is not consistent with how drugs are administered to patients.

In 2024, Montell [13], Douglas Green [12,16,17] (member of the NCCD), and their associates published comprehensive review/perspective articles in which they extensively discussed the mechanisms of cancer cell survival after engaging apoptosis and other pathways of regulated cell death. The main conclusion is the same as that drawn by the NCCD in their 2009 article [63], namely “… whereas previously we had always considered that there were ‘points of no return’ in any cell death pathway, we now realize that in many types of active, regulated cell death, this is not the case” [16].

It is noteworthy that Green and Montell will be co-chairing an upcoming forum (September 2025) to explore open questions surrounding regulated cell death, persister cells, and minimal residual disease in cancer [67]. The term “persistence” refers to a non-genetic state of tolerance towards radiation [59] or drugs [60] that contributes to dormancy, therapy resistance, and relapse. Hopefully the meeting will also include discussions on cell death-independent events (polyploidy, neosis, atavistic reprogramming) that also contribute to minimal residual diseases [3].

### 3.2. Cell Surface CD24 Expression in Anastatic Cancer Cells

Recently, Vasileva et al. [68] reported that the cell surface expression of CD24 is preferentially enriched in anastatic cancer (melanoma) cells that exhibit tumorigenic properties (also see the online article by Vijay Ulaganathan, one of the senior authors of this study [69]). Of note, even CD24-positive cancer cells that display various cell “death” indicators (trypan blue staining, annexin V staining, nuclear fragmentation, and detachment from the culture surface) are able to recover and form large colonies under 3D culture conditions [68]. (Doesn’t this call for revisiting thousands of articles that have used the terms “apoptosis” and “death” synonymously?) The discovery of CD24 as the cell surface marker of anastasis will be instrumental in advancing our understanding in the field of cell death pathways.

### 3.3. Tumor-Repopulating Property of Apoptotic Cancer Cells Is Not Limited to Anastasis

In addition to anastasis, apoptotic cancer cells are capable of promoting tumor repopulation through various other mechanisms (reviewed in [15]). These include the blebbishield emergency program (an anastasis-like phenomenon observed in cancer stem cells) [70,71], phoenix rising (caspase-3-mediated secretion of tumor-repopulating factors) [72,73], CASP3^+^ cell islands (densely populated apoptotic cells within an individual tumor that promote cancer cell survival) [74], nuclear expulsion (release of chromatin and associated proteins that promote metastasis) [75], and the reversal of senescence [76].

### 3.4. A Paradigm Shift Is Currently Occurring

As recently pointed out by Green [16,17], the realization that there is no point of no return in regulated cell death pathways amounts to a paradigm shift in the field. We have drawn a similar conclusion in a recent article entitled “Changing the Landscape of Solid Tumor Therapy from Apoptosis-Promoting to Apoptosis-Inhibiting Strategies” [15]. Referring to anastasis and other tumor-repopulating events shown in Figure 2, we stated that “Just thinking about all these complex processes makes one’s head spin, but unfortunately this is the established reality. We will be fooling ourselves if we continue to reduce this mind-numbing complexity to highly simplistic hypotheses, such as the notion that cancer cells treated with therapeutic agents either repair their genome and survive or die via apoptosis and other means” [15].

## 4. Is Overactive Anastasis a Hallmark of Cancer?

The realization that “apoptosis” is not the final destination of cancer cells undergoing regulated cell death raises an intriguing question: is deregulated anastasis a hallmark of cancer? In other words, is it possible that at least some forms of leukemia, for example, are associated with overactive anastasis? If so, is it possible to modulate anastasis (e.g., by pharmacologic approaches) to extend the life of leukemia patients? The discovery of the cell surface expression of CD24 as a marker of anastasis will enable us to address this and other outstanding questions in the context of complex diseases.

## 5. Who Would Disregard the Dark Side of Apoptosis in Cancer Therapy? Apparently Many!

The reversibility of apoptosis in mammalian cells was first reported by Geske et al. [77,78] in the early 2000s; it was rediscovered in 2009 [64], called anastasis in 2012 [4], and subsequently well documented for various cancerous and non-cancerous cell types. Furthermore, clinical studies demonstrating the dark side of apoptosis have been reported since the 1990s [25,29,32,33]. The purpose of reiterating these facts is to raise the following question: why is it that it took over two decades to regard apoptosis as a tumor-repopulating response, and by only a handful of cancer researchers? In other words, what are the reasons that most NCCD members (excluding Zhivotovsky and Green) still consider the various non-lethal outcomes after engaging apoptosis to be highly controversial and that “much more work must be done to verify or discard them” [37]? Possible reasons for this serious oversight have been discussed [3].

There are two ways of looking at this brief history of cell death research: sugarcoating and grim reality. By sugarcoating, I mean drawing the typical highly optimistic conclusions such as decades of hard work has brought us to here (i.e., the realization that apoptosis fuels the oncogenic process), and now we will be able to design novel strategies to combat cancer. The Nobel Prize Laureate William Kaelin has referred to such sugarcoating interpretations as “wishy-washy” (for details, see [3]). What percentage of cancer-related articles end with such wishy-washy conclusions? (Maybe >95%?)

The grim reality has recently been discussed in various publications, including books and online blogs, some of which were highlighted recently [15]. For example, on the topic of being continuously bombarded with rosy promises regarding the metaphor of the war on cancer, Dr. Otis Brawley (previous chief medical officer at the American Cancer Society) stated in a blog interview [79] something to the effect that many cancer researchers and oncologists are starting to believe their own fatally flawed hypotheses. (I am rephrasing this portion to avoid the use of strong words.) “The consequences are real—and they can be deadly. Patients and their families have bought into treatments that either don’t work, cost a fortune or cause life-threatening side effects” [79].

## 6. False Hypotheses That Continue to Hamper Progress in Cancer Research

The many hurdles in treating solid tumor malignancies are well documented and extensively discussed. According to Sarabjot Pabla [80], for example, the toughest hurdles include tumor heterogeneity, drug resistance, tumor microenvironment, metastasis, limited drug penetration, toxicity to normal tissues, tumor dormancy and relapse, and lack of predictive biomarkers. Decades of hard work has led us to identifying these hurdles. Tackling each one of them, however, will be a Herculean task, if possible at all.

Let us consider only one of these various hurdles, tumor heterogeneity, which has been the focus of the aforementioned Special Issues of MDPI publications [56] and the current commentary. The mechanisms that underline intratumor heterogeneity can be broadly categorized as cell-intrinsic and cell-extrinsic [81]. The intrinsic factors are associated with genome instability [81] and fall under the umbrella of genome chaos, a rapid and complex phenomenon of genome organization [82,83]. The best characterized intrinsic factors are listed in Figure 2. Extrinsic factors include intercellular communication and tumor’s blood supply (heterogeneity reflecting angiogenesis, hypoxia, oxidative stress, acidosis, etc.), as elegantly discussed by Sun and Yu [81] ten years ago.

It is unclear where all this ever-increasing complexity will lead in terms of treating solid tumors. What is crystal clear is that several false hypotheses, including the following four, have derailed cancer research for decades. Unfortunately, this trend continues.

➢**False hypothesis #1:** “In mammalian cells, the activation of executioner caspases occurs after the cells are committed to die”, as recently stated by the NCCD, based on reports that were published over 20 years ago (see [37]). In other words, apoptosis has a point of no return after which the cell fate (demise) is permanent. (This contradicts the earlier NCCD report [63]; see Section 3.1.) **Fact:** Now we know that there are no points of no return in various cell death pathways and that cancer cells can recover from late stages of apoptosis via anastasis, giving rise to aggressive variants.➢**False hypothesis #2:** In vitro “live/dead” assays generate clinically relevant information. To this end, in 2009 [63], the NCCD recommended that, in the absence of a clearly defined point of no return, a cell should be considered dead when the integrity of its plasma membrane is lost, as indicated by the incorporation of vital dyes such as propidium iodide (PI). Accordingly, the uptake versus exclusion of large, vital dyes (e.g., PI, trypan blue, acridine orange) has been ubiquitously used to discriminate dead cancer cells, regardless of the underlying cell death pathway. **Fact:** Loss of cell membrane integrity, detected by vital dye assays, is often transient. Thus, cancer cells can rapidly repair the injuries to their plasma membrane and survive. This fact was clarified in 2013 by Husmann [84] in a correspondence regarding the NCCD’s recommendations. Furthermore, as alluded to earlier, it has recently been demonstrated that cancer cells that are positive in trypan blue staining and PI staining assays are able to recover and proliferate [68,69].➢**False hypothesis #3:** Cancer cells evade apoptosis to survive anticancer treatment. (Why should cancer cells bother to evade a treacherous, tumor-repopulating response such as apoptosis?) **Fact:** High-grade cancers with poor prognosis contain relatively high levels of apoptotic cells. Thus, apoptosis promotes tumorigenesis rather than constituting a tumor suppression mechanism [9,14,15].➢**False hypothesis #4:** Identifying drugs that reinstate or promote apoptosis “could lead to disease regression or cures in patients with difficult-to-treat tumors” [38]. This outdated hypothesis still continues to be widely cited (e.g., [38]). **Fact:** Compelling clinical and preclinical studies outlined in the previous sections demonstrate that this highly simplistic hypothesis is not tenable for treating complex diseases such as cancer. Furthermore, in the context of treating solid tumors, there are bigger fish to fry than targeting the various regulated cell death pathways. Namely, the creation of PGCCs following anticancer treatment; these giants enter a state of dormancy after they are formed, undergo adaptation, and give rise to tumor-repopulating progeny via amitotic cell division (neosis) and other mechanisms (reviewed in, e.g., [3,85,86]).

## 7. Important Considerations When Assessing Cancer Cell Response to Therapeutic Agents

As we pointed out previously [86], the discovery of the DNA damage surveillance network (now referred to as the “DNA damage response”) in the 1990s led to a number of assumptions that have become almost “undisputable facts” (e.g., apoptosis, senescence, and “mitotic catastrophe” are permanent cell fates) and yet have proven to be untenable for most cell types, including solid tumor-derived cell lines. Some of these “hypotheses”, together with discoveries that need to be taken into consideration when assessing cancer cell response to therapeutic agents, have been extensively discussed. These include (i) the danger of relying on multiwell plate cell “viability” and in vitro colony formation assays for assessing cancer cell death following treatment with DNA-damaging agents [3,15,86]; (ii) the significance of the single-cell MTT assay, recently optimized by us [87], for evaluating the viability and metabolic activity of individual cancer cells. This assay is a powerful tool for distinguishing dead cancer cells and dying cells (e.g., exhibiting features of apoptosis, necroptosis, ferroptosis, etc.) that have the potential to undergo anastasis and give rise to tumor-repopulating progeny.

## 8. Conclusions

### 8.1. What Is Apoptosis?

We have been studying human cell responses to DNA-damaging agents for decades, even prior to the era of ATM, p53, and DNA damage response; since 1981, to be exact [88]. Despite this, only during the writing of this article did a common trend in cell death-related publications catch my attention. The NCCD [37] and many others (thousands) describe apoptosis as follows: activation of intrinsic or extrinsic signaling pathways of *apoptosis* inevitably leads to an end point called *apoptosis*. Triggering apoptosis leads to apoptosis? Meaning what? Death? Or a treacherous, tumor-repopulating process? The use of such vague statements is confusing and dangerously misleading.

### 8.2. Significance of Deintensification in Managing Solid Tumor Malignancies

The landscape of treating solid tumors is changing. Traditionally, the goal of radio/chemotherapy has been to stop or slow down the proliferation of cancerous cells, thus inhibiting tumor multiplication, invasion, and metastasis [89]. But this also results in severe side effects, which might include compromising the host immune system to fight cancer [90]. Furthermore, following the initial tumor response, anticancer treatment causes cancer cell dormancy that, in unknown time points, sometimes decades later, reactivates and triggers cancer relapse [59,61,85]. Therapy-induced cancer cell dormancy has been attributed to PGCCs, senescent cancer cells, radiation/drug-tolerant persister cancer cells, and quiescent cancer cells (Figure 2).

Intensive cytotoxic treatments or alternative anticancer strategies (e.g., synthetic “lethality”) have been used to destroy cancer cells via apoptosis and other regulated cell death routes. In the past decade, however, it has become increasingly evident that engaging regulated cell death is often accompanied by various prosurvival pathways, including anastasis, the natural cell recovery process that rescues cells from the brink of death. Intensive treatments also increase the incidence of various chaotic events (e.g., chromothripsis, polyploidy, amitosis, atavistic reprogramming), which propagate therapy-resistant and tumor-repopulating “monsters” (reviewed in, e.g., [3]).

The preclinical and clinical studies discussed herein underscore the significance of deintensification to minimize the occurrence of potentially life-threatening side effects as well as “treacherous” apoptosis and other tumor-repopulating events.

### 8.3. Take-Home Messages

The current article was prepared for the Special Issue of the *Int. J. Mol. Sci.*, entitled “The Cellular Response to DNA Damage: From DNA Repair to Polyploidy and Beyond” [91]. The main purpose of this Special Issue was to provide a comprehensive update on the growing complexity of molecular and cellular responses to DNA-damaging agents. The purpose was not to propose a novel therapeutic strategy because, in our opinion, this is currently impossible given that cellular complexity and heterogeneity within a single tumor has grown to an almost unimaginable degree ([3,15,86]; also see Figure 2).

As stated in the Editorial article published in 2023 [56], I trust that this Special Issue has provided sufficient discoveries for the reader to elaborate or debate on the main conclusion that we have reached [3]. Namely, “there is urgent need for designing preclinical anticancer assays (both cell-based and animal models) and treatment strategies that recapitulate the degree of complexity that exists within an individual solid tumor. Until then, at least in the near future, perhaps efforts of cancer researchers should be primarily directed towards prevention, rather than employing the same misleading preclinical assays and wishy-washy interpretations (to quote the Nobel Prize Laureate William Kaelin) with “novel” anticancer drugs and catchy names for treatment strategies (e.g., “synthetic lethality”) to expect different outcomes…” [3].

The recent discoveries highlighted herein provide further support for this conclusion. They also underscore the fact that a paradigm shift in the study of apoptosis and other cell death pathways is currently occurring [16]. Whereas previously many authors had assumed that there were points of no return in any pathway of regulated cell death, in the past decade, it has become evident that this is not the case [16].

## Figures and Tables

**Figure 1 ijms-26-01881-f001:**
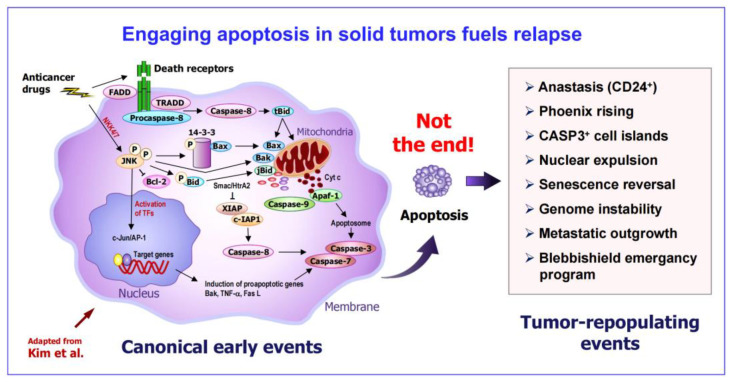
Early (canonical) and late (tumor-repopulating) events after engaging apoptosis in solid tumors. Early events: different stimuli trigger apoptosis, which involves the induction of proapoptotic genes, mitochondrial changes, caspase activation, loss of cell volume (shrinkage), and formation of membrane-bound apoptotic bodies (adapted from Kim et al. [41]). Late events: cancer cells can return from various stages of apoptosis, even after the formation of apoptotic bodies, via the homeostatic process of anastasis, often resulting in the emergence of more aggressive cancers. Several other prosurvival pathways are associated with apoptosis, as indicted. For details, see the text and [15].

**Figure 2 ijms-26-01881-f002:**
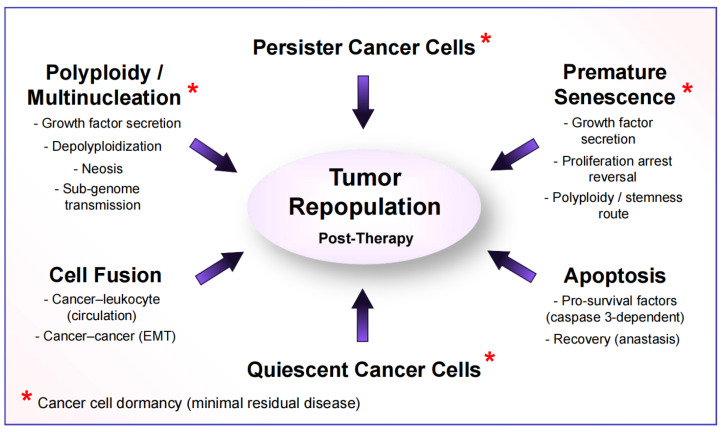
Therapy-induced responses in different subpopulations of cancer cells within a solid tumor that contribute to resistance and relapse. For details, see the text and [53].

**Figure 3 ijms-26-01881-f003:**
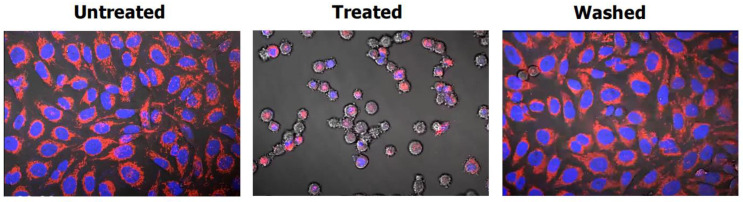
Representative images showing the induction of apoptosis (treated) followed by recovery (washed) in HeLa cervical carcinoma cells. The images are screenshots of a video presented online by Montell [66].

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
