# Peer review of "Anastasis and Other Apoptosis-Related Prosurvival Pathways Call for a Paradigm Shift in Oncology: Significance of Deintensification in Treating Solid Tumors"

_ijms, 2025, doi:10.3390/ijms26051881_

Round 1
Reviewer 1 Report
Comments and Suggestions for Authors
In the manuscript, the author provided detailed comments based on the analysis of several studies that pointed out the potential issues of apoptosis in anticancer studies. The author emphasized that the induced apoptosis by anticancer agents/treatments in cancer cells may not be the end of the cancer cells of solid tumors because some of these apoptotic/dying cancer cells may undergo anastasis. Then, tumor-repopulating events may be happened. The topic is interesting and important. Nonetheless, the author just highlighted the problem of apoptosis pathway for anticancer research. It needs some more positive comments on how to improve and address this known imperfect issue of apoptosis process in solid tumors. Can we conclude that apoptosis has no more value to be study further? I think we cannot deny that apoptosis is an important mechanism to cause cell death. How to make further improvement for this imperfect pathway is also important and interesting for readers in the field.
In addition, the author could give further comments on how long researcher should observe cell viability/MTT assays to measure whether anastasis happens. Apart from apoptosis, the author could comment on other better cell death pathways that researcher could study.
In general, the author reported very interesting cellular events and challenges in anticancer study. The manuscript may need more comments on how to improve and what alternatives could have. Readers may be more interested in these positive comments.
Author Response
Many thanks to both reviewers for taking the time to examine this commentary on an extremely complex and yet important topic, and for their positive comments. I trust that the following points clearly stand out: (i) we don’t know what cell death really is; (ii) Apoptosis is often (if not always) accompanied by anastasis and other prosurvival pathways (the screen shots in Figure 3 show that, shockingly, all cancer cells at late stages of apoptosis undergo anastasis); (iii) intratumor heterogeneity in terms of therapy resistance goes way beyond the apoptosis-anastasis response; (iv) This mind-numbing complexity calls for a paradigm shift in oncology to focus on strategies that minimize the occurrence of treacherous apoptosis and other tumor-repopulating events (e.g., therapy-induced cancer cell dormancy and reactivation).
In addition to completing the Figure 1 legend and correcting some typos, the following three revisions are made. All revisions are highlighted.
First, the text under the revised section 4 (“Is Overactive Anastasis a hallmark of Cancer?”) is not new, but it is moved from the end of the Conclusions section to the middle of the paper to stand out. I find this hypothesis to be the most important positive take home message of this commentary.
Second, section 7 under “Important Considerations When Assessing Cancer Cell Response to Therapeutic Agents” is new. It directs the interested readers to the indicated topics, including the single-cell MTT assay, which is a powerful tool for distinguishing dead cancer cells and dying cells that can recover and promote relapse.
Third, a new sub-section (#8.3) is added to the conclusions section which highlights the take home messages of the current commentary as well as the corresponding Special Issue.
Reviewer 2 Report
Comments and Suggestions for Authors
-

Author Response

(The authors gave the same response as above.)

Round 2
Reviewer 1 Report
Comments and Suggestions for Authors
The revised manuscript could be recommended for publication in its current form.